# Solvent Extraction Studies of Copper from a Heap Leach Liquor Using Mextral 5640H

**Mostafa Hosseinzadeh** [1,2,*]**, Jochen Petersen** [1] **and Asghar Azizi** [3]

1 Hydrometallurgy Research Group, Chemical Engineering Department, University of Cape Town, Cape Town 7700, South Africa
2 Research and Development Division, Zagros Mes Sazan (ZMS) Copper Company, Saveh 39141-39141, Iran
3 Faculty of Mining, Petroleum and Geophysics, Shahrood University of Technology, Shahrood 36199-95161, Iran
* Correspondence: hssmos002@myuct.ac.za

**Abstract:** In this study the extractive capability of Mextral 5640H was investigated for extraction of copper from a heap leach liquor. In this regard, the influence of parameters such as pH (0.2–2.8), extractant concentration in kerosene diluent (2.5%–10% *v/v*), temperature (25–70 °C), contact time (0–300 s), stirring speed (100–1200 rpm), phase ratio (O/A) (0.6–1.8) and Cu initial concentration (0.5–2 g/L) in the leach liquor were examined and optimized. The findings demonstrated that the Mextral 5640H extractant had a very high efficiency and selectivity in copper extraction from the leachate. 98.17% Cu, with less than 0.5% of Fe and Mn, were extracted at pH 1.6, 10% (*v/v*) Mextral 5640H concentration, ambient temperature (25 °C), 400 rpm stirring speed, 2 min contact time and an O/A phase ratio of 1:1. Under equilibrium conditions it was found that one mol of Cu is extracted by 7 mol of Mextral 5640H. Additionally, analysis using a McCabe–Thiele diagram suggests a two-stage extraction to reach the maximum extraction of copper (99.5%) from the leachate at operational condition using industrial mixer-settlers. Furthermore, a thermodynamic study was conducted, and the measured values of $\Delta H = 15.13$ kJ/mol, $\Delta G = -6.95$ kJ/mol and $\Delta S = 74.10$ J/mol/K indicate an endothermic, spontaneous nature and high affinity of copper extraction.

**Keywords:** solvent extraction; heap leach liquor; Mextral 5640H; copper; McCabe–Thiele diagram; thermodynamics





## 1. Introduction

Hydrometallurgical processes are applied as an efficient technique to extract many metals from ores and raw materials for which the use of conventional routes (especially high-temperature processes) is not economical or is problematic. The main advantage of this technology is lower cost, relatively simple operation, easy control and they are more environmentally acceptable [1–4]. These processes consist of multiple steps, including the dissolution (leaching) of desirable metals or components from a solid material such as ore, tailings or secondary resources, concentration and purification of the leach liquor (usually by solvent extraction) and generation of the final product (precipitation or electro-winning of metal) [5–7].

The possibility to up-concentrate metals from dilute solutions opens the opportunity of using low-grade mineral resources and industrial waste containing small quantities of the target metals as source materials [8–10].

There are two distinct categories of extractants targeting copper called ketoximes and aldoximes. Ketoximes are relatively strong extractants that separate copper at a pH above 1.8 with good efficiency [11,12]. Aldoximes are improved extractants and are particularly efficient at extracting copper at lower pH and temperature [13,14]. The properties of the compounds containing both these two types of extractants depend on the percentage of each in the mixture. In recent years, a vast number of studies have been performed on

the capability of different oxime solvents to extract copper from leach liquors such as CuPRO MEX-3302 [6], LIX 973N [8], LIX-54 [15], LIX 84-I and LIX 622N [16], LIX612N [17], LIX984 [18], Acorga M5640 [19,20], which were all shown to be highly efficient in extracting copper under commercial conditions. For instance, Reddy et al. [8] applied LIX 84 and LIX 973N for the separation of copper from a sulphate leach liquor of synthetic Cu–Ni–Co–Fe matte and found that the extraction efficiency of copper was 83% and 95% using LIX 84 and LIX 973N, respectively, with the values of Ni, Co and Fe each being <2 mg/L in loaded organic. Soeezi et al. [18] indicated that more than 85% copper could be selectively extracted with 10% (*v/v*) LIX 984 concentration diluted in kerosene. They reported that the greatest extraction percentage was obtained within 3 min at phase ratio (O/A) of 1:1, mixing rate of 500 rpm and 28 °C temperature. It was also found that the stripping stage efficiency of Cu was 93% using sulfuric acid at the phase ratio of O/A = 1 and 28 °C. Very recently, Ruiz et al. [21] applied two extractant groups including pure ketoxime (LIX 84-IC) and nonylaldoxime (LIX 860N-IC) for the selective separation of copper from a sulfate–chloride solution. They demonstrated that the stripping of copper from LIX 860N-IC was extremely difficult even at high concentrations of sulfuric acid.

Despite many studies performed on developing efficient and cost-effective reagents to selectively extract copper, the extractive behavior of Mextral 5640H has not yet been reported. We initially characterize the extraction behavior of copper via Mextral 5640H in a sulfate leaching liquor. In this regard, the impact of key parameters, i.e., pH, temperature, extractant time, initial copper concentrations, extractant concentration, phase ratio, stirring speed and extraction mechanisms are analyzed in detail and the mechanism of extraction process are examined.

## 2. Experimental Procedure

### 2.1. Materials

The stock aqueous solution used in this research was prepared from a sample taken from a sulfuric acid heap leaching process of copper ore at Zagros Mes Sazan (ZMS) Co., Saveh, Iran. This solution was analyzed by an atomic absorption spectrophotometer (Agilent 240FS AA, Santa Clara, CA, USA). Table 1 presents the chemical analysis of the pregnant leach solution (PLS). As can be seen, the Cu content in the leach liquor is about 2.07 g/L. For investigating the effect of copper initial concentration, Cu content was varied from 0.5 to 2 g/L.

**Table 1.** Chemical analysis of PLS.

| Element | Al | As | Ca | Co | Cu | Fe |
|---|---|---|---|---|---|---|
| Content (mg/L) | 2030 | 109 | 4590 | 101 | 2070 | 2680 |
| Element | K | Mg | Mn | Ni | Pb | Zn |
| Content (mg/L) | 261 | 1730 | 3740 | 19.8 | 0.770 | 202 |

Mextral5640H (5-Nonyl-2-hydroxy-benzaldoxime), modified aldoxime, 98.5% in purity were supplied by Kopper Chemical Industry Corp., Ltd., Chongqing, China. It is marketed as a highly efficient copper extractant and has the advantages of high Cu/Fe selectivity, stable performance, low solubility in water (<0.1%), good phase separation and less entrainment. It is widely used for the extraction of copper from mine acidic leaching solution, recovery of copper from industrial wastewater and extraction of copper from electroplating sludge and it is suitable for PLS with medium to low pH. The organic phase in the solvent extraction tests was prepared using Mextral 5640H as extractant and kerosene as diluent to reach the desired concentration. In order to adjust the pH solution in aqueous phase, sulfuric acid ($H_2SO_4$, 95%–98%) and ammonium hydroxide ($NH_4OH$, 25%), provided from Merck GmbH (Darmstadt, Germany), were utilized in the experiments. All other chemical reagents used during the experiments were of analytical grade, and all solutions at specified concentrations were prepared or diluted by deionized water.

### 2.2. Extraction Experiments

To perform the extraction experiments, the leach liquor pH was first adjusted to the desired value of 0.2–2.8 (determined with a HANNA Instruments pH meter, Model HI5221) and the organic phase was prepared by diluting Mextral 5640H in kerosene in a predetermined concentration of 2.5%–10% (*v*/*v*). Thereafter, the initial solvent extraction tests to investigate the effect of pH and extractant concentration on Cu separation were done with 50 mL volumes each of aqueous and organic phases (except those for the McCabe–Thiele diagram) for 30 min at an agitating rate of 400 rpm using magnetic stirrer (except for study of effect of the stirring speed) to reach an extraction equilibrium state at room temperature (25 °C). Experiments to test the temperature effect were performed under magnetic stirring and heating by hotplate in the range of 25–70 °C. During the tests pH and temperature control was carried out, keeping pH between 0 and 3 at different temperatures (25 ± 2 °C, 40 ± 2 °C, 55 ± 2 °C, 70 ± 2 °C). After finishing the extraction process, the two phases were settled a separating funnel, and copper content in the aqueous phase was measured by atomic absorption spectrophotometry (AAS). A mass balance was conducted to calculate the amount of copper extracted into the organic phase, and Equation (1) was applied to determine the extraction efficiency (E) of copper as follows:

$$E = \frac{Cu_f - Cu_r}{Cu_f} \times 100 \tag{1}$$

where $Cu_f$ and $Cu_r$ denote the concentration of Cu in the leach liquor and raffinate, respectively.

In order to demonstrate the effect of pH on selectivity of Cu over Fe and Mn, a separation factor (β) was determined as a selectivity parameter of copper solvent extraction from leach liquor. In this regard, a greater value of β indicates a more selective separation. The separation factor (β) of two metals A and B is expressed as:

$$\beta = D_A / D_B \tag{2}$$

where the distribution coefficient, D, of metal A is describes as:

$$D = [A]_{org} / [A]_{aq} \tag{3}$$

where $[A]_{org}$ and $[A]_{aq}$ are the metal ion concentration in the loaded organic and aqueous phase, respectively, after each extraction.

### 3. Results and Discussion

#### 3.1. Effect of pH and Extractant Concentration

To investigate the effect of the extractant concentration on the amount of copper extracted from the leach liquor, experiments considering 2.5%, 5%, 7.5% and 10% (*v*/*v*) Mextral 5640H in kerosene were conducted at the phase ratio (O/A) 1:1 at ambient temperature (25 °C) for 30 min (Figure 1). The pH values used for the experiments were in the range of 0.2–2.8. As can be seen from Figure 1, with increasing concentration of the extractant and increasing pH, the extraction percentage increases. It was also found that the isotherms shift to the left with the increment of Mextral 5640H concentration, indicating an increase in the extraction power with reagent strength, at least at lower pH values. In fact, with increasing the extractant concentration more Mextral molecules are dissolved into the aqueous phase, therefore more $Cu^{2+}$ ions can be complexed and thus transferred from aqueous solution back to the organic phase. This results in an overall increase in the mass transfer between two phases even at lower pHs, as shown in the Figure 1, while at high pHs, it remains relatively unchanged. This behavior may be due to the formation of solids or colloidal particles at high pH levels for oxime-based extractants [22]. It was also observed that an amount of crud accumulated during the tests, which increased with increasing extractant concentration at higher pH values [23].

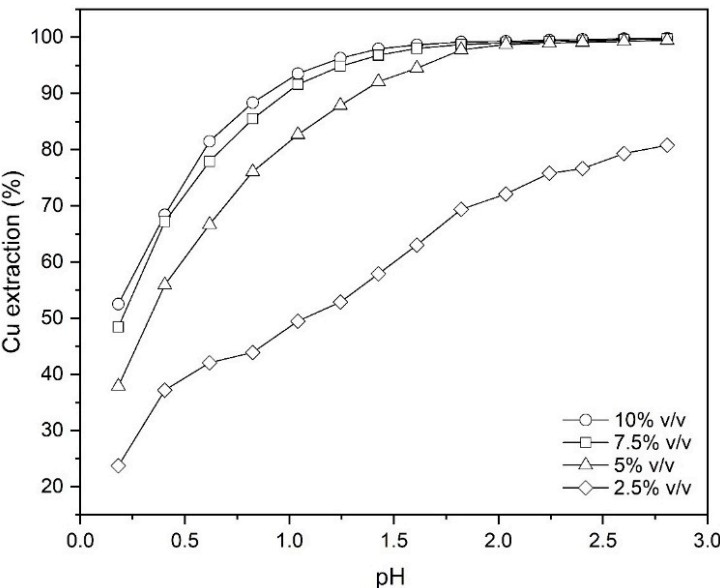

**Figure 1.** The effect of Mextral 5640H and pH on the Cu extraction (T = 25 °C, O/A = 1 and t = 30 min).

Liu et al. [23] explained that the reason for the increase in the amount of crud with the increase in the amount of extractant can possibly be due to the increase of extractant concentration in that the poly-components accumulate considerably in the organic phase, resulting in an increase in the amount of crud by visual observation. In addition, it was seen that the time required for the separation of the two phases increased at the higher pH values due to the formation of a third phase during the separation process (Figure 2).

Comparing the data of copper extraction at two concentrations of 7.5% and 10% (*v/v*), it was found that there is not much difference between them. Therefore, economically, using the extractant at a concentration of 7.5% (*v/v*) is optimal in terms of extractive performance, but since a part of the extractant is always removed from the system in the aqueous raffinate because of problems such as crud and dissolution in the aqueous phase, a concentration of 10% (*v/v*) should be used with some margin of safety.

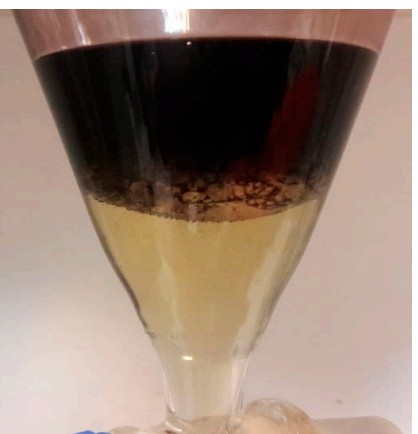

**Figure 2.** Separation of aqueous and organic phases (pH = 2.6, O/A = 1, Mextral 5640H = 10% (*v/v*), T = 25 °C).

Figure 3 illustrates the dependence of extraction of copper on the pH. In addition to the copper, the extraction efficiency of main impurities, including manganese (Mn) and iron (Fe), are displayed as an indication of the selectivity of extractant performance. According to Figure 3, the extraction efficiency of copper was significantly promoted from 52.5% to

99.8% by varying the pH from 0.2 to 2.8. Therefore, Mextral 5640H has a high selectivity relative to the major impurities (i.e., Mn and Fe) in copper PLS; the extraction percentages of iron and manganese in the selected pH range were between 0%–0.2%—within the analytical error.

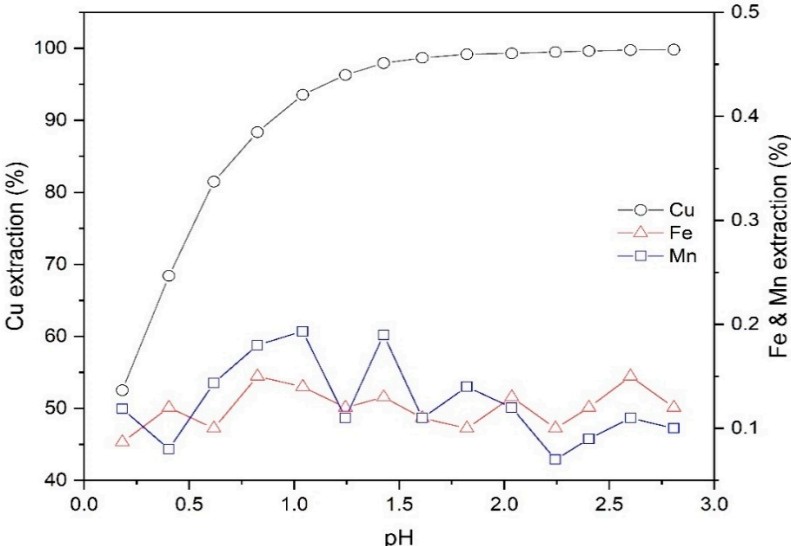

**Figure 3.** The effect of pH on the extraction of Cu, Fe and Mn (Mextral 5640H = 10% (*v/v*), T = 25 °C, O/A = 1 and t = 30 min).

The Cu separation factor towards Fe and Mn at different values of pH is shown in Table 2. As shown, the selectivity of Cu over Fe and Mn improves with increasing pH. Based on the data plotted in Figure 3, since over 98% copper was extracted at pH 1.6 and given the values of $\beta_{Cu/Fe}$ and $\beta_{Cu/Mn}$ at this pH are considerable, the extraction of Cu from leach solution was further investigated using pH 1.6.

**Table 2.** The values of separation factors at different pH.

| pH | $D_{Cu}$ | $D_{Fe}$ | $D_{Mn}$ | $\beta_{Cu/Fe}$ | $\beta_{Cu/Mn}$ |
|---|---|---|---|---|---|
| 0.2 | 1 | 0.095 | 0.135 | 12 | 8 |
| 0.4 | 2 | 0.136 | 0.087 | 16 | 25 |
| 0.6 | 4 | 0.111 | 0.168 | 40 | 26 |
| 0.8 | 8 | 0.176 | 0.220 | 43 | 35 |
| 1.0 | 14 | 0.163 | 0.240 | 89 | 60 |
| 1.2 | 26 | 0.136 | 0.124 | 191 | 210 |
| 1.4 | 47 | 0.149 | 0.235 | 318 | 202 |
| 1.6 | 74 | 0.124 | 0.124 | 600 | 600 |
| 1.8 | 120 | 0.111 | 0.163 | 1080 | 737 |
| 2.0 | 141 | 0.149 | 0.136 | 945 | 1036 |
| 2.2 | 195 | 0.111 | 0.075 | 1757 | 2593 |
| 2.4 | 273 | 0.136 | 0.099 | 2003 | 2762 |
| 2.6 | 409 | 0.176 | 0.124 | 2317 | 3308 |
| 2.8 | 551 | 0.136 | 0.111 | 4041 | 4959 |

Since PLS from a commercial operation was used during the SX tests, it should be noted that the $Mn^{2+}$ ions in the leach liquor can be transferred by the organic phase to an electrolyte solution in EW where they are chiefly converted to $Mn^{3+}$ ions and to manganese dioxide. During the stripping process, these oxidised components can be delivered back to the organic phase, where they can potentially oxidise some organic components and thus destroy them. This oxidation behaviour of manganese species can cause weak phase separation and create stable emulsions during the solvent extraction

process [24,25]. Therefore, regarding the results from the Figure 3 and Table 2, Mextral 5640H was shown to have a very high selectivity of Cu over Mn, which in turn, minimizes the transmission of Mn from the aqueous solution to the organic phase.

Given the negligible extraction of these impurities, the extraction of Fe and Mn was not investigated further in the study of other parameters.

### 3.2. Effect of Contact Time

To investigate copper extraction at different times, some tests were conducted over a range of 0–300 s at various pH values using 10% (*v/v*) Mextral 5640H in kerosene, O/A ratio = 1:1 and ambient temperature (25 °C) (Figure 4). As shown on the graph, after about 100 s, more than 97% of copper was extracted at pH = 1.6. It can be concluded that with increasing pH, the extraction equilibrium is reached more rapidly. For example, to achieve equilibrium extraction at pH = 0.8 takes about 180s, but it only takes 80 s for that extraction to be completed at pH = 2.4. As shown in Figure 4, at pH = 1.6 after about 120 s, prolonging the contact time had no further impact on the extraction. Therefore, the contact time tests showed that the extraction equilibrium was achieved in approximately 2 min. To minimize any errors the extraction time used during the rest of the experiments was chosen to be 10 min, except for temperature tests which were done in 30 min. It was also decided not to develop a kinetic model for this work as extraction was always sufficiently fast to reach equilibrium to not influence the design of an industrial contactor.

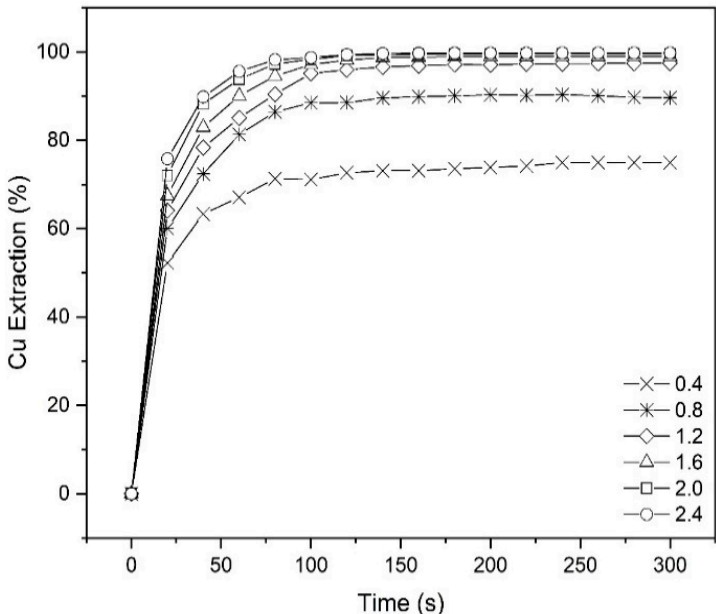

**Figure 4.** The effect of contact time on copper extraction percentage at different pHs (Mextral 5640H = 10% (*v/v*), O/A = 1, T= 25 °C).

### 3.3. Effect of Stirring Speed

Figure 5 characterizes the effect of stirring speed in the solvent extraction process of copper from the leach liquor. It is clear that the extraction percentage of copper strongly increases from 34.5 to 98.1% with an increase of stirring rate from 100 to 400 rpm. At higher velocities, due to the production of smaller size of aqueous and organic droplets, the contact surface between the particles increases, and as a result, the mass transfer rate from the aqueous phase to the organic phase increases [26]. As shown in Figure 5, increasing the stirring speed beyond 400 rpm had no further effect on copper extraction.

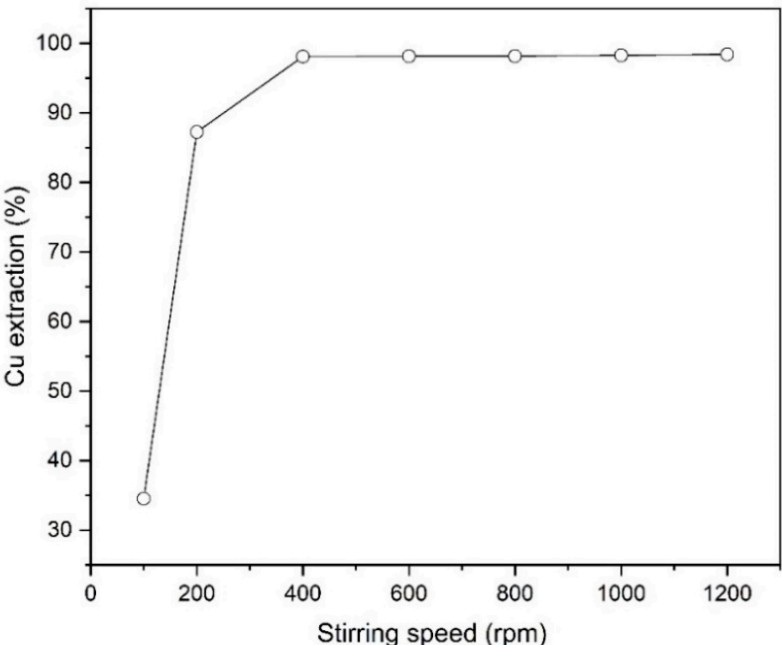

**Figure 5.** The effect of stirring speed on Cu extraction (Mextral 5640H = 10% (*v/v*), pH = 1.6, T = 25 °C, t = 10 min and O/A = 1).

*3.4. Effect of Phase Ratio*

　　The impact of phase ratio on Cu extraction was examined at different amounts of O/A ratios (0.6, 0.8, 1, 1.2, 1.4, 1.6 and 1.8) using 10% (*v/v*) Mextral 5640H at initial pH of 1.6 and ambient temperature (25 °C). Figure 6 shows that increasing the ratio of O/A from 0.6 to 1.8, the extraction percentage of copper increases from 88.4% to 99.8%. In addition, it was found that at the higher O/A ratios of 1.8 the formation of a third phase at the interface was not observed.

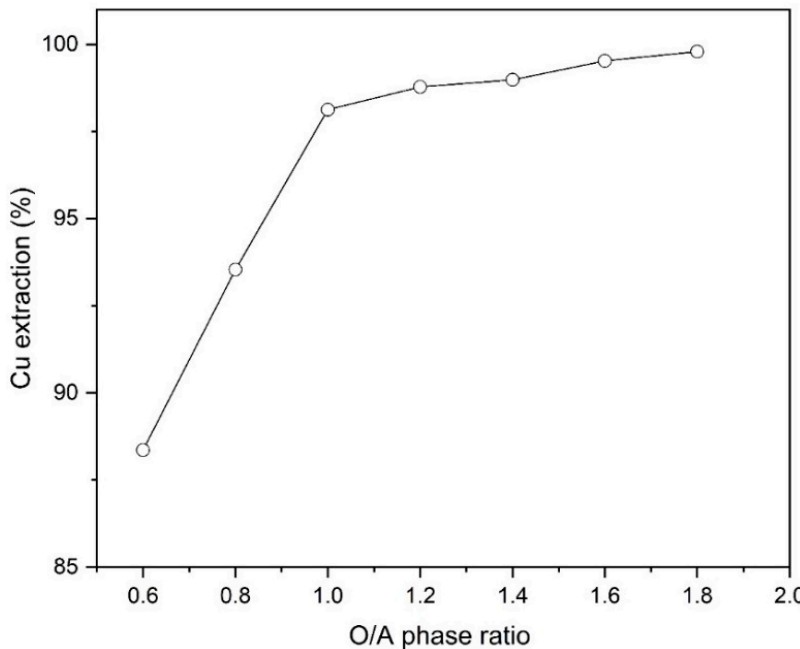

**Figure 6.** The effect of different phase ratios (O/A) on Cu extraction percentage (Mextral 5640H = 10% (*v/v*), pH = 1.6, T = 25 °C, t = 10 min and rpm = 400).

### 3.5. Effect of Cu Concentration in Leach Liquor

As shown in Figure 7, at a lower pH the amount of copper which could be loaded by the Mextral 5640H reagent decreases with an increase of the initial concentration of Cu in leach liquor.

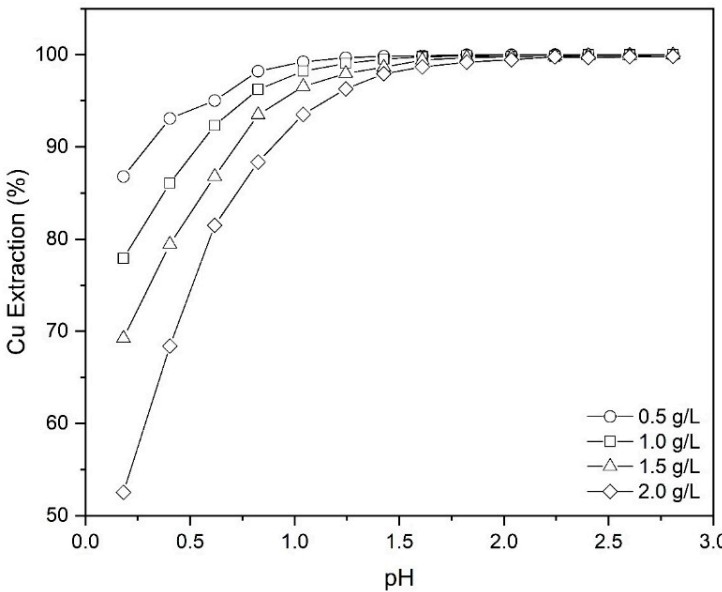

**Figure 7.** The effect of initial Cu concentration on copper extraction at different pHs (Mextral 5640H = 10% (*v/v*), T = 25 °C, t = 10 and O/A = 1 and rpm = 400).

To determine the number of extraction stages required at the chosen volume phase ratio (O/A = 1) of Mextral 5640H, an extraction isotherm (McCabe–Thiele diagram) was plotted between the amount of copper extracted by the solvent and copper left unextracted in the raffinate at different O/A ratios at pH = 1.6. As shown in Figure 8, a two-stage counter-current extraction would achieve an extraction efficiency of 99.5% Cu at pH 1.6, 10% (*v/v*) Mextral 5640H in kerosene, stirring speed 400 rpm, contact time 10 min and room temperature (25 °C).

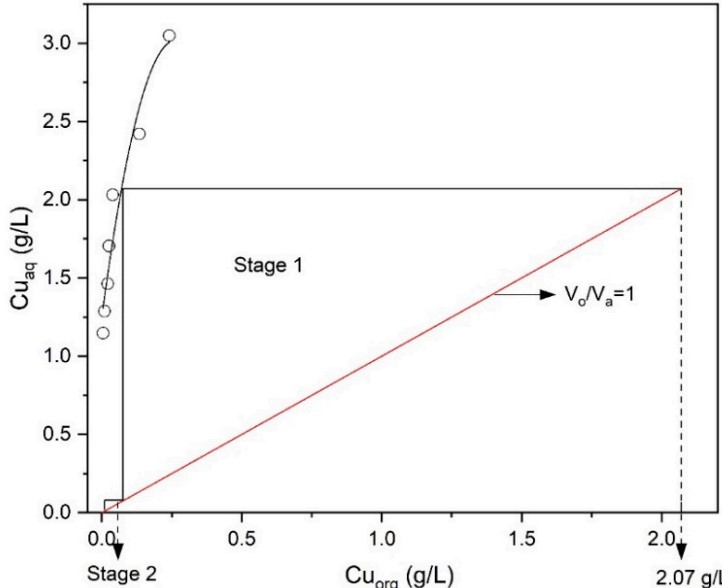

**Figure 8.** McCabe–Thiele diagram for extraction of Cu (Mextral 5640H = 10% (*v/v*), pH = 1.6, T = 25 °C, t = 10 min, rpm = 400).

### 3.6. Extraction Mechanism

The mechanism of extraction process was studied to identify and understand the extraction behavior of Cu from the leach liquor using Mextral 5640H. The general extraction reaction of divalent metal ions, such as $Cu^{2+}$, from sulfate solutions using Mextral 5640H can be written as follows:

$$Cu^{2+}_{(aq)} + nRH_{(org)} \Leftrightarrow R_2Cu \cdot (RH)_{n-2(org)} + 2H^+_{(aq)} \qquad (4)$$

where RH depicts extractant and $R_2Cu \cdot (RH)_{n-2}$ refers to the organic phase loaded with $Cu^{2+}$ ions. For Equation (4), the equilibrium constant (k) can be written as:

$$k = \frac{\left[R_2Cu \cdot (RH)_{n-2}\right]_{org} \left[H^+\right]^2_{aq}}{\left[Cu^{2+}\right]_{aq} [RH]^n_{org}} \qquad (5)$$

where the square brackets indicate solution concentration of the various species. Strictly speaking, activities should be used, but the activity coefficients of all species were considered constant within the range of conditions tested and, so assumed, lumped into the equilibrium constant and separation factors.

Since $D_{Cu} = \frac{\sum[Cu]_{org}}{\sum[Cu]_{aq}} = \frac{\left[R_2Cu \cdot (RH)_{n-2}\right]_{org}}{\left[Cu^{2+}\right]_{aq}}$, so Equation (5) can be represented as:

$$k = D_{cu} \frac{\left[H^+\right]^2_{aq}}{[RH]^n_{org}} \qquad (6)$$

Taking the log of both the sides of above Equation (6) yields:

$$\log D_{Cu} - n \log[RH]_{org} = \log k + 2pH \qquad (7)$$

in which the Mextral 5640H concentration in the equilibrium phase can be determined as follows:

$$[RH]_{org} = [RH]_o - n \times \left[\overline{M}\right]_{org} \qquad (8)$$

Equation (7) can be analyzed through a linear regression by plotting $\log D_{Cu} - n \log[RH]_{org}$ vs. pH for different values of n for the experimental data as shown in Figure 9. The linear plot has the offset logk and the slope needs to match the value of 2. This is achieved by n = 7. This result implies that each copper cation complexed with two dissociated RH will be surrounded and stabilized by a further 5 RH molecules in the organic phase.

Thus, for Mextral 5640H Equation (4) could be rewritten as follows:

$$Cu^{2+}_{(aq)} + 7RH_{(org)} \Leftrightarrow R_2Cu \cdot (RH)_{5(org)} + 2H^+_{(aq)} \qquad (9)$$

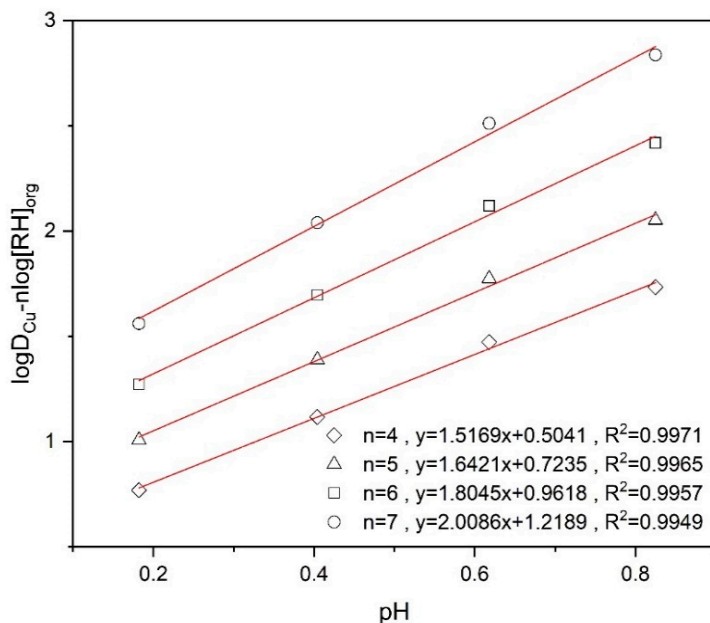

**Figure 9.** $\log D_{Cu} - n\log[RH]_{org}$ versus pH at different n values using 10% ($v/v$) Mextral5640H.

### 3.7. Effect of Temperature and Thermodynamic Study

The temperature studies on the extraction of copper from the leach liquor using Mextral 5640H were done at different temperatures, 298, 313, 328 and 343 K at the pH ranging from 0.2 to 2.8, 10% ($v/v$) Mextral 5640H concentration, O/A = 1:1 and mixing time of 30 min. As shown in Figure 10, with increasing the pH to values higher than 1.5, there are no noticeable changes in the extraction efficiency of copper, whereas it is obvious that at lower pH, for example 0.2, there is a 14% difference in the magnitude of copper extraction between 298 and 343 K.

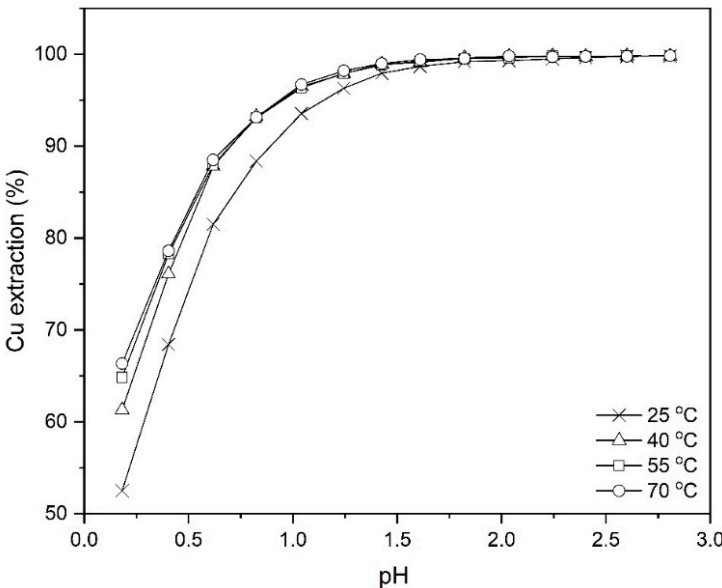

**Figure 10.** Effect of temperature on Cu extraction at different pH values (Mextral 5640H = 10% ($v/v$), O/A = 1:1, t = 30 min, rpm = 400).

Thermodynamic investigations were also carried out to recognize the feasibility of the extraction process of copper using Mextral 5640H. In this regard, van't Hoff relationships

were used to measure the thermodynamic parameters (ΔH, ΔS and ΔG) according to the following equations:

$$\log D_{cu} = \frac{-\Delta H}{2.303R}\frac{1}{T} + C \tag{10}$$

$$\Delta G = -RT\ln K_{ex} \tag{11}$$

$$\Delta S = \frac{\Delta H - \Delta G}{T} \tag{12}$$

where T, C, R, ΔH, ΔG and ΔS stand for the temperature (K), a constant for a solution and the universal gas constant (8.314 J/mol/K), enthalpy change, Gibbs free energy change, and entropy change, respectively. Figure 11, which represents the linear relationships between log D and 1000/T, indicates how to calculate the ΔH using Equation (10) for the extraction of copper. After that, Gibbs free energy (ΔG) was determined from Equation (11) with the equilibrium constant ($K_{ex}$) and the entropy change (ΔS) was determined by Equation (12) at 298 K (Table 3).

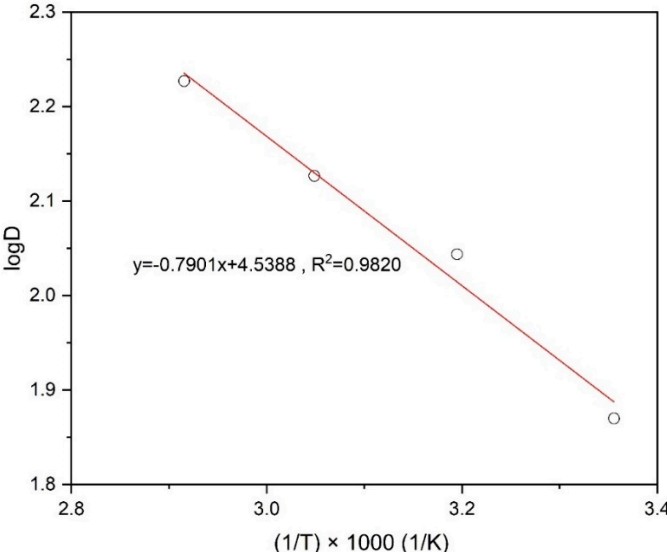

**Figure 11.** Plot of log D against 1000/T for the extraction of Cu using Mextral 5640H.

The values calculated for the above thermodynamic parameters are shown in Table 3. Considering the results presented in Table 3, it is clear that the extraction of copper from the leach liquor by Mextral 5640H is an endothermic process (ΔH = 15.13 kJ/mol) and the extraction reaction is a spontaneous process with increasing temperature (ΔH > 0 and ΔS > 0).

**Table 3.** Thermodynamic parameters obtained for Cu extraction using Mextral 5640H.

| Thermodynamic Parameters | Values |
|---|---|
| ΔH kJ/mol | 15.1 |
| ΔG kJ/mol | −6.95 |
| ΔS J/mol/K | 74.1 |

Finally, the extractive capability of Mextral 5640H was compared with a number of other widely used commercial solvents such as Acorga M5640, LIX group and CuPRO MEX-3302 as shown in Table 4. It can be seen that the Mextral 5640H reagent has an extremely high selectivity and extractive ability (98.17% Cu with a value of impurities (Fe and Mn) less than 0.5%) compared to the other solvents under highly acidic conditions.

**Table 4.** The present work in comparison with some previous studies.

| Extractant | Leaching Solution | Extraction Conditions | Result of Extraction | Reference |
|---|---|---|---|---|
| CuPRO MEX-3302 | Sulfate leach solution containing Cu, Fe and Mn | Extractant concentration = 5% ($v/v$), O/A phase ratio = 1:1, contact time = 3 min, pH = 2.1 and temperature = 25 °C | 95% Cu with Fe and Mn content less than 2% | [6] |
| LIX 84 and LIX 973N | Sulphate leach liquors of synthetic Cu–Ni–Co–Fe matte | Extractant concentration = 40% ($v/v$) and O/A phase ratio = 1:1 | 83% Cu by LIX 84 and 95% by LIX 973N with residual content less than 2 mg/L. | [8] |
| LIX 984 | synthetic and industrial solutions | Extractant concentration = 10% ($v/v$), O/A phase ratio = 1:1, contact time = 3 min, pH = 2–3 and temperature = 28 °C | 85% and 77% of Cu from synthetic and industrial solutions, respectively. | [18] |
| LIX 984 N | Bioleaching solution containing | Extractant concentration = 20% ($v/v$), O/A phase ratio = 1:1, contact time = 4 min, pH = 1.5, stirring rate = 400 rpm and temperature = 25 °C | 96% Cu at two stages counter-current extraction | [27] |
| LIX 84I and LIX 622N | Waste heat boiler dust leach liquor | Extractant concentration = 30% ($v/v$), O/A phase ratio = 1:1, contact time = 3 min, pH = 3.5 and room temperature | About 80.8% Cu with LIX 622N and 69.4% Cu with LIX 84I. | [28] |
| Mextral 5640H | Sulfate leach liquor | Extractant concentration = 10% ($v/v$), O/A phase ratio = 1:1, contact time = 2 min, pH = 1.6, stirring rate = 400 rpm and temperature = 25 °C | 98.17% Cu with a value of Fe and Mn < 0.5%. | Present work |

In the context of Table 4, it can be seen that Mextral 5640H has shown a much better performance than LIX in terms of extractant consumption, pH and extraction time under comparable conditions. It could achieve over 98% Cu extraction using 10% of extractant at pH = 1.6 in 2 min, while the extraction efficiency of Fe and Mn at this pH was less than 0.2%. Therefore, using this extractant, it is possible to obtain a high degree of separation of copper from the leach liquor at lower values of pH without significant extraction of other impurities. However, in comparison to the results achieved from previous studies using Acorga M5640 [19,20] is of particular interest as the chemical composition of both reagents is very similar with 5-Nonyl-2-hydroxy-benzaldoxime the key active reagent.

Nozari and Azizi [20] showed that in order to extract copper from a leach solution with initial Cu and Fe concentration of 2083 mg/L and 637 mg/L, respectively, using 10% ($v/v$) Acorga M5640 and experimental conditions of phase ratio O/A = 1 and 25 °C temperature, the leach liquor needed to reach pH = 2.5 to achieve an extraction percentage of 92.3% Cu from solution, while using the same conditions Mextral 5640H could achieve 98.2% copper at pH = 1.6. Under the given operational conditions, Mextral 5640H extraction tests achieved equilibrium conditions in about 2 min from the start of the process while a contact time of 20 min was required for the Acorga M5640 tests. Therefore, extraction equilibrium was obtained much more quickly for Mextral 5460H.

Furthermore, using the extractant concentration (10% ($v/v$)), O/A = 1 and T = 25 °C, over 93% copper was extracted at pH = 1 by Mextral 5640H while at the same conditions this amount was only about 40% for Acorga M5640[20] which indicates better suitability of Mextral 5640H for highly acidic media. Regarding iron and manganese as the most important impurities in the copper leach solutions, the studies clearly showed that at the same conditions (extractant concentration at 10% ($v/v$), 25 °C temperature and O/A ratio of 1/1 at pH = 1.6) the separation factor of Cu/Fe reached 600 using Mextral 5640H while this amount was only 7.61 for Acorga M5640 (at pH = 1.5), which consequently translates to a high purity of copper in the loaded organic phase [20].

Similarly, Wang et al. [19] showed that using a leach liquor with initial copper and iron concentration of 6166 mg/L and 57,500 mg/L, respectively, extraction by 16% ($v/v$) Acorga M5640 at pH = 1.1, phase ratio O/A = 1/1, contact time 3 min and 25 °C temperature, removed 88.6% of copper from the leach solution and a Cu/Fe separation factor of only around 355 was achieved.

The superior performance of Mextral 5640H over Acorga M5640 needs to be understood in terms of the extractant composition. According to the manufacturer, Mextral 5640H is a modified and improved reagent with some fundamental changes in the molecule

structure, in order to extract more selectively and be stripped more easily than LIX reagents, but the exact nature of these modifications remains proprietary. The association of Cu ions with an additional five molecules of extractant, as shown in the present study, could potentially offer a hint at the formation of a particularly stable macro-molecule with much reduced surface charge density, which may enhance its rapid re-absorption into the organic phase. This would explain both the faster kinetics and superior performance observed for Mextral 5640H.

## 4. Conclusions

This study aimed to investigate the extraction behavior of copper from a heap leaching liquor with the impurities (Fe, Mn) using Mextral 5640H as an extractant developed specifically for copper plants. The results indicated that upwards of 98.7% Cu could be extracted rapidly from sulfate leach liquor under the optimized condition. It was also found that Mextral 5640H had very high separation factors and selectivity in copper extraction at pH $\geq$ 1 with minimal co-extraction of Fe and less than 0.5% for Mn. Thermodynamic studies showed that the extraction process of Cu is an endothermic and naturally spontaneous reaction. In addition, the mechanism of extraction was examined, and it was determined that about seven moles extractant were needed to remove one mole Cu from the leach liquor, of which two are assumed to be chemically complexed with the copper ion, whereas the remaining five are coordinated around the complex. According to the McCabe–Thiele diagram drawn for the system, a two-stage extraction process would be needed to attain a copper extraction efficiency of 99.5%. Mextral 5640H outperforms the chemically similar Acorga M5640, likely to be due to structural modifications enhancing a more rapid adsorption of the formed Cu complex into the organic phase.

**Author Contributions:** Conceptualization/methodology/investigation/writing—original draft preparation, M.H.; writing-review and editing, J.P. and A.A. All authors have read and agreed to the published version of the manuscript.

**Funding:** This research received no external funding.

**Data Availability Statement:** Not applicable.

**Acknowledgments:** The authors would like to express their gratitude and appreciation to Zagros Mes Sazan (ZMS) Co. for supplying the samples and equipment in this study.

**Conflicts of Interest:** The authors declare no potential conflict of interest.

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
