# Peer review of "Solvent Extraction Studies of Copper from a Heap Leach Liquor Using Mextral 5640H"

_minerals, doi:10.3390/min12101322_

Round 1

Reviewer 1 Report

This paper discusses the extraction performance of copper using Mextral 5640H (5-Nonyl-2-hydroxy-benzaldoxime). The main contribution of the paper is the systematic extraction tests and thermodynamic parameters of Mextral 5640H. Both datas and references are well organized in this paper. However, I would like to see some discussion of the extraction selectivity mechanism of Mextral 5640H and an in-depth exploration of why Mextral 5640H achieved better Cu recovery than other extractants. I recommend that this paper be accepted after revision.

Author Response

Reviewer# 1 comment:

This paper discusses the extraction performance of copper using Mextral 5640H (5-Nonyl-2-hydroxy-benzaldoxime). The main contribution of the paper is the systematic extraction tests and thermodynamic parameters of Mextral 5640H. Both data and references are well organized in this paper. However, I would like to see some discussion of the extraction selectivity mechanism of Mextral 5640H and an in-depth exploration of why Mextral 5640H achieved better Cu recovery than other extractants. I recommend that this paper be accepted after revision.

Response:

We have revised the discussion section (lines 320-370) to focus especially on the comparison with Acorga M5640, which is chemically similar with the same core reagent 5-Nonyl-2-hydroxy-benzaldoxime. The exact nature of the modifications effected in Mextral 5640H are proprietary, but given the novel observation of 7 molecules associated with each Cu instead of the expected 2, points at the formation of a macromolecule which is more easily adsorbed back into the organic phase explaining both the faster kinetics and higher extent of the extraction with Mextral 5640H over Acorga M5640.

Reviewer 2 Report

You find here enclosed some comments and suggestions about your paper:

- Table 1: Results obtained should be presented using three significant digits (not decimals), i.e. 2030; 109; 4590; ...., 19,8; 0.770; 202...

- Line 121: Change Ph by pH

- Line 155, Figure 2: change phrases  to phases

- Line 250: I think that this sentence should be included after Eq. 5, not before

- Equation 6??

- Line 266: Mextral?

- Table 3: three significant digits, 15.1; -6.95; 74.1

- Lines 303-337: this paragraph shoulb be rewritten for comprension purposes

Author Response

Reviewer# 2 comments

- Table 1: Results obtained should be presented using three significant digits (not decimals), i.e. 2030; 109; 4590; ...., 19,8; 0.770; 202...

- Line 121: Change Ph by pH

- Line 155, Figure 2: change phrases  to phases

- Line 250: I think that this sentence should be included after Eq. 5, not before

- Equation 6??

- Line 266: Mextral?

- Table 3: three significant digits, 15.1; -6.95; 74.1

Response:

All the comments and suggestions above were considered and addressed on the paper.

- Lines 303-337: this paragraph should be rewritten for comprehension purposes.

Response

In line with the suggestions from reviewer #1, this section has been rewritten and a more careful comparison between Mextral 5640H and Acorga M5640 has been presented. The superior and more rapid performance of Mextral has been tentatively explained on the basis of the formation of larger complex molecules (through the association of each Cu with 7 extractant molecules) which are more readily re-adsorbed into the organic phase.